# Hybrid Structure of a ZnO Nanowire Array on a PVDF Nanofiber Membrane/Nylon Mesh for use in Smart Filters: Photoconductive PM Filters

**Dong Hee Kang** , **Na Kyong Kim and Hyun Wook Kang** *

Department of Mechanical Engineering, Chonnam National University, 77 Yongbong-ro, Buk-gu, Gwangju 61186, Korea; kdh05010@gmail.com (D.H.K.); naky0607@gmail.com (N.K.K.)
* Correspondence: kanghw@chonnam.ac.kr; Tel.: +82-62-530-1662

**Abstract:** A nanofiber membrane with a high surface-to-volume ratio has advantages in applications such as those used for particulate matter filtration and gas detection. To maximize the potentials of the membrane structure, recent research has been attempted to control nanofiber geometries. In this paper, surface modification of a nanofiber membrane with a metal/ceramic nanostructure is performed to improve multi-functional filter performance, enhancing fine particle filtration and toxic gas absorption. Here, a smart filter is fabricated by electrospinning polyvinylidene difluoride (PVDF) nanofiber onto a nylon mesh and hydrothermal synthesis of ZnO nanoparticles onto a nanowire array on a PVDF nanofiber surface. On the ZnO nanowires–PVDF nanofiber layer filter, the pressure difference ($\Delta P$ = 4.13 kPa) is higher than the pure PVDF nanofiber layer. However, the filtration efficiency is 94.3% for a 0.3 µm particle size, which is higher than that of other sizes. Additionally, a ZnO nanowire array with high density on a PVDF nanofiber layer affects sensitivity ($S$ = 39.37), with high resolution. The photocurrent characteristics of a smart filter have the potential for a photo-assisted redox reaction to detect toxic polar molecules in continuous airflow in real-time in indoor environments.

**Keywords:** smart filter; particulate matter; hybrid structure; electrospinning; pvdf; hydrothermal synthesis; zinc oxide; photo conductivity; quality factor

## 1. Introduction

Micro/nanostructure-based thin film fabrication technology are used in a wide range of industries and research fields for integration systems and device applications, and are related to energy applications such as flexible electronics, soft photonics, supercapacitors, and surface modification [1–10]. The characteristics of the high surface-area-to-volume ratio of the hierarchical structure dramatically enhances performance to achieve their applications. For examples, these characteristics provide a large number of active sites that can be utilized for ion transfer in batteries [3], detecting gas with high sensitivity and fast response in sensors [5], and controlling cell behavior using a high-density functional group on the film surface [7].

A fiber-based porous film fabrication using electrospinning technology is one of the thin film fabrication methods for the mass production for filters in industries [8,11]. This paper endeavors to increase the particulate matter (PM) filtration efficiency. In the nanofiber (NF) fabrication process, the viscosity of the polymer solution and intensity of the applied voltage is controlled for the spider-web-like fibrous structure [12]. A nanofiber net filter structure shows a high filtration efficiency of over 99.9%, with a low pressure drop (<100 Pa) [13]. Furthermore, in the post-process of the filter fabrication, triboelectric charges are applied to the NF filter surface to help capture PM below a diameter of 0.3 µm [14]. In this way, PM filtration is enhanced by controlling geometry during the NF manufacturing step or by applying continuous external static forces to the filter surface.

Including basic PM filtration performance, the multifunctional filter is designed to provide additional performance using a hybrid structure [15–22]. Modulating the nanofiber surface (surface functionalization and metal oxide combination) improves the content of the active sites, which can be effective in absorbing volatile organic compounds (VOCs), sulfur dioxide, and carbon monoxide. Souzandeh et al. presented surface functionalization of gelatin nano-fabrics on a paper towel to capture PM and toxic gaseous molecules. The protein chain of gelatin NFs has a molecular functional group that captures gasses such as sulfur dioxide and carbon monoxide, as well as volatile organic compounds [18]. Additionally, the $MnO_2$ embedded PE/PP fiber filters demonstrate air purification for PM 2.5 filtration with excellent catalytic activity for formaldehyde [19]. The nanofiber membrane filter, which is a modulated nanofiber surface functionalization or a metal oxide combined structure, shows superior filtration efficiency and provides gas capture characteristics.

In this research, a smart filter is fabricated onto the structure of zinc oxide (ZnO) nanowire (NW) arrays on the polyvinylidene difluoride (PVDF) NF layer. ZnO, a semiconductor with large excitation binding energy (60 meV) at room temperature, is a promising material for a wide range of applications, such as photodetectors or chemical sensors [23,24]. In addition, the wurtzite crystalline structure of the ZnO is beneficial to grow nanowire geometry at a low-temperature under 100 °C. With a low-temperature hydrothermal synthesis method, it is possible to fabricate a hierarchical structure on a polymer membrane without thermal damages [2]. The ZnO NWs-PVDF NF layer performance is analyzed by the photo conductivity (in the current-voltage curve and the current-time curve) and the PM filtration characteristics (in the pressure difference, the PM filtration efficiency, and the quality factor). These characteristics demonstrate a multifunctional smart filter application capable of both PM filtration and toxic polar gas detection.

## 2. Materials and Methods

### 2.1. PVDF Nanofiber Membrane Fabrication on Nylon Mesh

An electrospinning technique and hydrothermal synthesis were used for PVDF NF layer fabrication on a nylon mesh substrate and ZnO NW forming on a PVDF NF surface, respectively. First, before the electrospinning process, Poly(vinylidene difluoride) (PVDF, Mw ~ 534,000, Sigma-Aldrich, St. Louis, MO, USA) powder was dissolved with 13 wt% into the mixed solvents acetone (99.5%, Sigma-Aldrich, USA): n,n-dimethylformamide (DMF, 99.8%, Sigma-Aldrich, USA) (7:3) at 50 °C for 72 h using an ultrasonic bath (AJC2010, OMAX, Korea) with the highest cavitation intensity to prepare a homogeneous PVDF solution. After sonication, the PVDF solution was stirred at 40 °C for one week. It was then allowed to cool down for one hour at room temperature before the electrospinning process began.

In the preparation process for electrospinning, as shown in Figure 1a, a commercial nylon (woven Nylon 66A mesh, Flon Ind., Tokyo, Japan.) mesh layer was used as a substrate for stacking the PVDF NF layer. The mesh layer has a low resistance to airflow in filter applications and serves as a supporting structure to reduce the physical stresses of the PVDF NF layer. Nylon mesh substrate was placed on an electrical grounding aluminum foil layer. An acrylic plate that would induce the electric fields, with a hole (Diameter: 70 mm) in its center, was placed over the nylon mesh to selectively fabricate the PVDF NF layer in the electrospinning process. A voltage source was applied at +15 kV to the metal needle tip, as shown in Figure 1b. Positively charged PVDF droplets elongated into nano-fibrous shapes to create an equilibrium state between the electrostatic repulsion and the surface tension of the solution at the metal needle tip. The PVDF solution in the electric field was ejected onto the nylon mesh substrate, 150 mm away from the metal needle tip at a flow rate of 30 μL·min$^{-1}$ for 600 s. The PVDF solution concentration was optimized to fabricate uniform nanofiber geometry for the fabrication of the filter membrane with a low-pressure drop. (Figures S1 and S2, presented in the Supplementary Material) After the electrospinning process, the PVDF NF layer on the nylon mesh was dried in air to remove residual solvents at room temperature for 24 h.

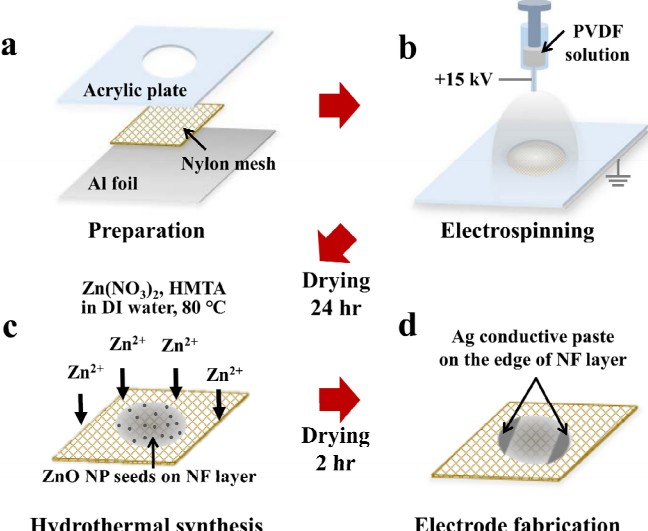

**Figure 1.** Schematics of the electrospinning, hydrothermal synthesis, and electrode fabrication process: (**a**) Preparation for electrospinning; (**b**) Fabrication of polyvinylidene difluoride nanofiber layer onto the nylon mesh layer; (**c**) Hydrothermal synthesis for ZnO nanowire fabrication after ZnO nanoparticle seeding on PVDF NF layer; (**d**) Electrode coating on smart filter using Ag paste for electrical properties analysis.

### 2.2. ZnO Nanowire Fabrication on the PVDF Nanofiber Membrane

Low-temperature (<10 °C) hydrothermal synthesis technology forms NWs through crystallization of ZnO on the surface of PVDF NFs. Before the hydrothermal synthesis, ZnO nanoparticles (NPs) and ZnO precursor solution were prepared. First, the ZnO NPs were synthesized in ethanol (>99.5%) solution containing 0.030 M sodium hydroxide and 0.010 M zinc acetate at 60 °C for 2 h. Then, 0.025 M zinc nitrate hexahydrate, 0.025 M hexamethylenetetramine, and 0.001 M polyethylenimine were dissolved in deionized water at 95 °C for one hour for the preparation of the ZnO precursor solution. All chemicals for ZnO NW fabrication were purchased from Sigma-Aldrich (USA) and did not require additional processes for purification. A ZnO NPs seed layer coating was performed by dripping and drying the ZnO NP-dispersed ethanol solution onto the PVDF NF layer. Finally, this was immersed in the ZnO precursor solution at 80 °C for 12 h, as shown in Figure 1c. After hydrothermal synthesis, the PVDF NF layer surface was washed with ethanol to clean out debris and was then dried at room temperature for 2 h.

### 2.3. Measuring Photoconductivity of the ZnO NWs-PVDF NF Layer

On the ZnO NWs-PVDF NF layer, a silver conductive paste (Ag paste, Elcoat P-100) was used to fabricate electrodes in parallel to the edge of the membrane, as shown in Figure 1d. The silver paste was cured at room temperature for 12 h. The photoconductive characteristics of the ZnO NWs-PVDF NF layer was analyzed through light-induced electron emission.

## 3. Results

### 3.1. Geometrical Analysis

The scanning electron microscope (SEM) images of the PVDF NF layer, with and without ZnO NWs, are shown in Figure 2; the diameter distributions are shown in Figure 2c,f. The geometry of the PVDF NF layer shows an arbitrary orientation, with a Gaussian distribution of the fiber diameter. On the surface of the PVDF NF layer, the average fiber diameter is 260.5 ± 148.8 nm. The NF surface shows a smooth and high aspect ratio without spindle-shaped beads. The ZnO NWs, through hydrothermal synthesis, grew in a vertical direction along the PVDF NF surface, as seen in Figure 2d,e. The ZnO NWs on the PVDF NF surface show a hierarchical structure with an average diameter of 1158.5 ± 482.2 nm.

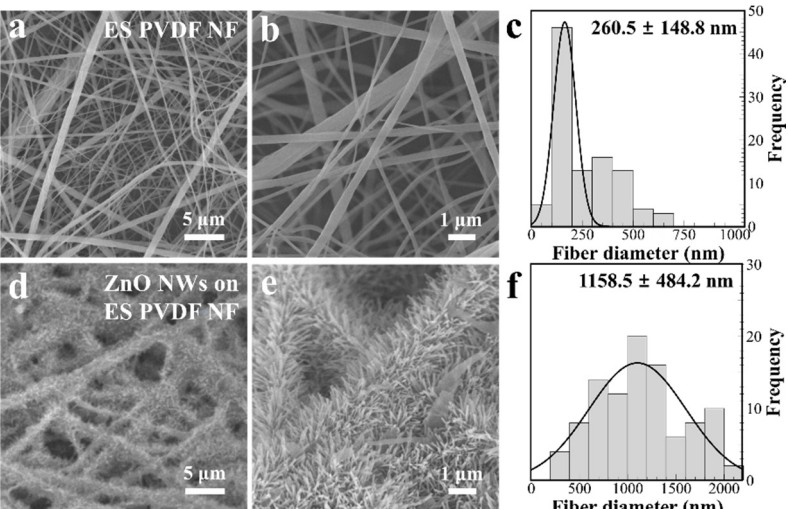

**Figure 2.** Scanning electron microscope (SEM) images showing surface morphology analysis of (**a**,**b**) PVDF NF layer and (**d**,**e**) ZnO NW on the PVDF NF layer. (**c**,**f**) Average diameter and distribution analysis of the NFs.

In Figure 3, the X-ray diffraction (XRD) result shows the identical crystallinity of the PVDF NF layer, regardless of the ZnO NWs fabrication, because the hydrothermal process is not affecting the chemical reaction on the PVDF in low-temperature synthesis. The dominant β-phase crystalline structure ($2\theta = 20.33°$) of PVDF NF was caused by mechanical stretching and electrical poling to molecular chains during the electrospinning process. The other intensity peaks ($2\theta = 31.69°$ (100), $34.39°$ (002), $36.19°$ (101)) in the XRD show the high crystallinity of the ZnO NWs by the wurtzite structure.

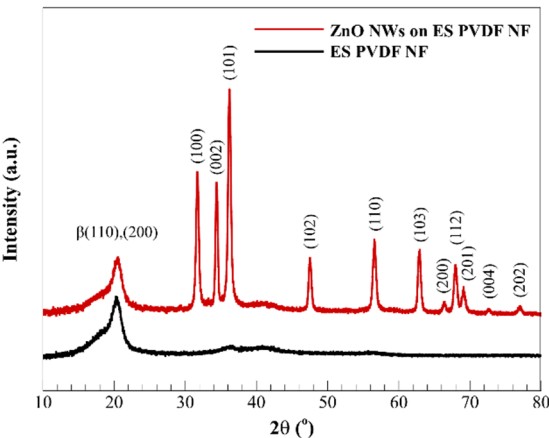

**Figure 3.** X-ray diffraction analysis results of electrospun PVDF NF layer, with and without ZnO NWs.

A solar simulator was used as a light source of 1/4 sun intensity (24 mW cm$^{-2}$). The current-voltage (I-V) and current-time (I-t) characteristics of the ZnO NWs-PVDF NF layer under the solar simulator were measured using a Keithley 2400 source meter (Tektronix, Beaverton, OR, USA).

### 3.2. Surface Energy Characteristics

The static contact angle was measured on the films of the PVDF NF layer with and without ZnO NWs using a sessile deionized water (DI water) droplet (4 μL) for surface energy characteristics analysis, as shown in Figure 4. PVDF is a fluoropolymer whose higher fluorine contents increase hydrophobicity of the membrane [25]. The surface morphology of the NF layer induces the increased roughness, which enhances the hydrophobicity.

Contact angle variation by surface roughness is expressed by the Cassie–Baxter model, as in Equation (1):

$$\cos\theta_{CB} = r_f \Phi_S \cos\theta + \Phi_S - 1 \tag{1}$$

where $r_f$ is the roughness ratio of the film and $\Phi_S$ is the ratio of the adjacent solid surface to the liquid interface; $\theta$ is the equilibrium contact angle from a derivation of Young's equation.

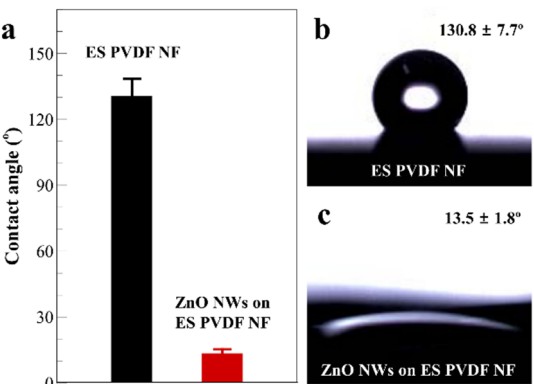

**Figure 4.** (**a**) The static contact angle measurement and analysis on the film surface of (**b**) an PVDF NF layer and (**c**) an PVDF NF layer with ZnO NWs.

In contrast, the ZnO NWs structure on the NF surface converts the PVDF NF layer to hydrophilicity. The hydrophilic property of the intrinsic ZnO is changed to superhydrophilicity, depending on the increasing surface roughness, which contacts with a water droplet. The contact angle changed by the surface roughness of the hydrophilic layer is expressed in Wenzel's model, as in Equation (2):

$$\cos\theta_w = r\cos\theta \tag{2}$$

where $r$ is the true area of the NF surface to the apparent contact area. The increased surface roughness of the ZnO surface on the PVDF NF layer improves the surface-to-volume ratio, whose structure is favorable for charge transportation for gas detections on the ZnO NWs in continuous airflow. The sequential images in Figure 5, using a high-speed camera, show a DI water droplet dripping onto the ZnO NWs-PVDF NF layer. After dripping onto film, the droplet is absorbed beneath the film surface within 5 s. The hygroscopic film characteristics shows that the ZnO NWs on the PVDF NF surface attract a water droplet due to the capillary force between the NW structures, while simultaneously trapping the liquid in the pores of the PVDF NF layer. Depending on the surface energy of the base material in contact with the droplet, a transition to the Cassie–Baxter state and Wenzel state occurs as the surface roughness increases, which induces reinforcement of the hydrophobicity and hydrophilicity, respectively [26].

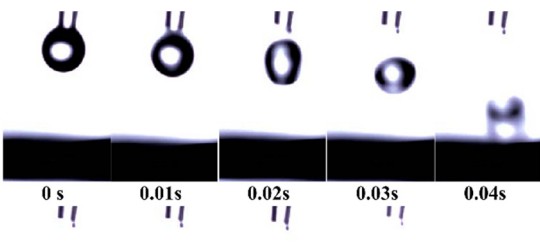

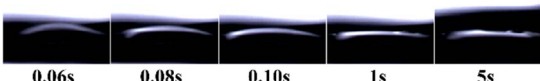

**Figure 5.** High-speed image sequence of DI water droplet dripping onto the ZnO NWs-PVDF NF layer surface.

### 3.3. Photoconductive Characteristics

The effectiveness of the photoconductivity of the ZnO NWs-PVDF NF layer structure is confirmed by I-V and I-t characteristics curves. As shown in Figure 6a, the I-V characteristic curve is analyzed by measuring the DC voltage sweep from −10 to 10 V with an increment of 0.01V. The I-V curves indicate the conductivity of the film under dark and light source radiation in atmospheric conditions. The resistance ratio ($R_{off}/R_{on}$) of the ZnO NWs-PVDF NF layer is 40.9. Light response performance has affected the density of the ZnO NWs on the layer [27]. The sensitivity ($S$) of the ZnO NWs-PVDF NF layer is expressed as Equation (3) [28]:

$$S(\%) = \frac{I_{on} - I_{off}}{I_{off}} \cdot 100 \qquad (3)$$

where $S$ is 39.37, from which I-V curve data are used to determine the sensitivity as a 1% truncated average. Compared with the present photodetectors, based on ZnO nanostructures [29], the ZnO NWs-PVDF NF layer demonstrates the applicability of a light-assisted polar gas molecule detector at room temperature. The time-dependent photocurrent response shown in Figure 6b is the I-t characteristic curve of ZnO NWs on a PVDF NF layer at 1 V DC bias for 1000 s under continuous light source radiation and the dark condition. Initially, photoconductivity response increases rapidly up to 0.028 μA for 30.9 s. After a maximum current point, the current shows a decay power curve, y = a·xb (a = 0.0624, b = −0.2184), through the curve fitting.

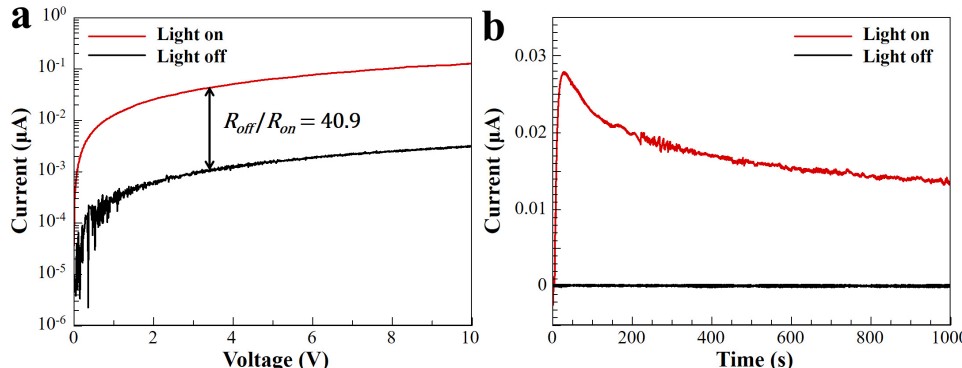

**Figure 6.** (**a**) I-V characteristics and (**b**) I-t characteristics (at 1V bias) of ZnO NWs on electrospun PVDF NF under continuous light source of solar simulator (at 24 mW·cm$^{-2}$) and dark conditions.

### 3.4. PM Filtration Characteristics

The PM filtration characteristics of a PVDF NF layer based on a nylon mesh substrate were analyzed. Figure 7 shows the pressure difference, filtration efficiency, and quality

factor on the layers of the PVDF NF surface, with and without ZnO NWs. The pressure difference and filtration efficiency were measured in a filter test chamber on the upstream and downstream sides of the filter. The filter layer was inserted into a test chamber with a diameter of 50 mm. Intense combustion and dilution were performed at the upstream side of the filter chamber, where PM concentration was maintained to the level of 150 $\mu$g·m$^{-3}$. At the downstream side of the filter chamber, a face velocity (24.8 cm·s$^{-1}$) on the filter layer was controlled through vacuum pump suction for 0.5 h.

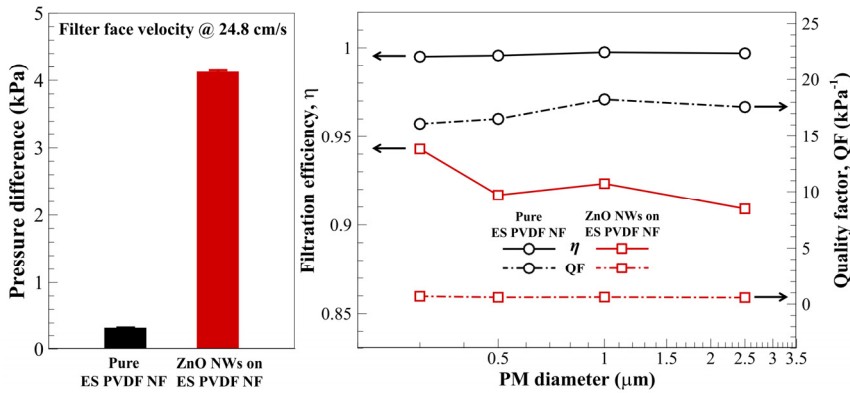

**Figure 7.** PM Filtration performance analysis of electrospun PVDF NF layer, with and without ZnO NWs, via measurement of pressure difference ($\Delta P$), filtration efficiency ($\eta$), and quality factor (QF).

The $\Delta P$ of the ZnO NWs-PVDF NF layer shows 4.129 ± 0.0175 kPa, which is 12.5 times higher than a PVDF NF layer (0.329 ± 0.0040 kPa). However, the filtration efficiency ($\eta$, %) of the ZnO NWs-PVDF NF layer was over 90.9%, regardless of the PM size. The filtration efficiency of 0.3 $\mu$m size particle was 94.3%, which is higher than that of other particle sizes conditions. The quality factor (QF), which shows correlation between pressure difference and filtration efficiency, is expressed as Equation (4):

$$QF = -\frac{ln(1-\eta)}{\Delta P} \quad (4)$$

In the ZnO NWs-PVDF NF layer, QF is 0.69 kPa$^{-1}$ (@ 0.3 PM), which is lower than the PVDF NF (16.09 kPa$^{-1}$ @ 0.3 PM) layer due to the high pressure drop. The PVDF NF layer has superior QF due to high filtration efficiency (99.4% @ 0.3 PM) with low pressure difference. The ZnO NWs-PVDF NF layer is required to ensure a decreasing pressure difference of the filter for multifunctional filter applications.

In summary, a high-density ZnO NW structures on a PVDF NF layer (a ZnO NWs-PVDF NF layer filter) shows high filtration efficiency (94.3%) for fine particulate matter (<0.3 $\mu$m) under 24.8 cm s$^{-1}$ face velocity. The nylon mesh layer is used as a supporting layer for the ZnO NWs-PVDF NF filter layer, which maintains NF structure without geometric deformation due to the low-pressure differential variation (< ±0.5%). The ZnO NW array is fabricated with high crystallinity on the NF surface without chemical reaction to PVDF via low temperature (80 °C) hydrothermal synthesis. The high surface-area-to-volume ratio of the ZnO NWs-PVDF NF layer has possibilities as a photoelectric generator, which has been demonstrated through the sensitivity ($S$) analysis of the filter under visible light radiation at ambient temperature. The filter characteristics analysis has shown that the structure of the ZnO NWs-PVDF NF layer filter has remaining limitations for immediate commercial use as a multifunctional filter because the hydrophilic filter layer, when exposed to a humid environment, is susceptible to pressure drop characteristics. However, these challenges in the filter performance can be overcome by improving the air permeability of the filter. In particular, electrospinning conditions allow easy control of the film geometry controls, such as changes in NF diameter and film thickness, which can reduce the pressure differences. In hydrothermal synthesis, the aspect ratio of NW structures can be simply

controlled to improve the functionality of filters with photocurrent properties. For further applications using a hierarchical structure based on a high porosity layer, it is possible to fabricate a high-performance sensor for gas detection in the continuous fluid flow environment.

## 4. Conclusions

A multifunctional smart filter to perform PM capture and photo-detection was fabricated by electrospinning for PVDF NF layer formation and low-temperature hydrothermal synthesis for ZnO NW growth. Filter layers, with and without ZnO NWs, were analyzed according to geometry, surface energy, and filtration performances. In addition, the ZnO NWs-PVDF NF layer was analyzed regarding its photo-conductive characteristics under 1/4 sun intensity light radiation and dark condition. After ZnO NW growth on the PVDF NF layer, the surface energy is converted from hydrophobicity (130.1°) to hydrophilicity (13.5°), and the ZnO NW structure on a porous NF layer has hygroscopic characteristics due to the high surface energy of the intrinsic ZnO and capillary forces. The high-density ZnO NWs are sufficient for light detection, with 39.37 sensitivity in the DC voltage sweep under the light source on and off conditions. Regarding the PM filtration performance, the quality factor in the PVDF NF layer with ZnO NWs is lower than the pure PVDF NF layer due to the high-pressure difference (>1 kPa) because of the dense NW structure in the pores of the NF substrate. However, improvement of the functionality of the multifunctional filter is possible through geometry control of the NW (density per area, length, and diameter) and NF (film thickness, fiber surface and sectional geometry, diameter distribution, and average diameter) structure in the fabrication process. The photo-current property of a high-density ZnO NW array with a hybrid NF membrane/mesh filter structure has exhibited great potential for detecting toxic gases, such as VOCs (e.g., aldehydes, alcohols, ketones), and also demonstrates fine PM filtration.

**Supplementary Materials:** The following are available online at https://www.mdpi.com/article/10.3390/app11178006/s1, Figure S1: The SEM images of the PVDF nanofiber membrane depending on the PVDF concentration, Figure S2: Pressure difference of the PVDF nanofiber membrane depending on the PVDF concentration.

**Author Contributions:** Conceptualization, D.H.K., N.K.K. and H.W.K.; methodology, D.H.K.; formal analysis, D.H.K. and N.K.K.; investigation, D.H.K. and N.K.K.; writing—original draft preparation, D.H.K.; writing—review and editing, D.H.K., N.K.K. and H.W.K.; supervision, H.W.K.; funding acquisition, D.H.K. and H.W.K. All authors have read and agreed to the published version of the manuscript.

**Funding:** This research was supported by the Basic Science Research Program through the National Research Foundation of Korea (NRF) funded by the Ministry of Education (NRF-2019R1A6A3A13096916), and the X-mind Corps program of National Research Foundation of Korea (NRF) funded by the Ministry of Science, ICT (No. 2021H1D8A1109673).

**Institutional Review Board Statement:** Not applicable.

**Informed Consent Statement:** Not applicable.

**Data Availability Statement:** Not applicable.

**Conflicts of Interest:** There are no conflict of interest to declare.

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
