# Peer review of "Hybrid Structure of a ZnO Nanowire Array on a PVDF Nanofiber Membrane/Nylon Mesh for use in Smart Filters: Photoconductive PM Filters"

_applsci, doi:10.3390/app11178006_

Round 1

Reviewer 1 Report

Comments

  1. The previous studies on this topic, some cited in the manuscript, have already shown the beneficial effects of including ZnO nanowires on different fibers for filtration application. Thus, the authors should clarify the novelty of their work compared to the state-of-the-art literature.

  1. Authors should include the other studies, which have investigated similar materials and parameters.

  1. The authors claimed that it is possible to improve the functionality of the filter through tuning different parameters. This is a too general claim, whcih should be supported by investigating some of the mentioned parameters.

  1. It would be better to include the durability of the ZnO nanostructures on the PVDF surfaces (e.g., under vigorous washing conditions).

  1. The authors should comment and better clarify the effects of the thickness/size of the applied ZnO layer/nanowires on the final applications.

Author Response

Manuscript ID: applsci-1348520

"Hybrid Structure of ZnO Nanowire Array on PVDF Nanofiber Membrane/Nylon Mesh for Smart Filter: Photoconductive PM Filter" by Dong Hee Kang, Na Kyong Kim, Hyun Wook Kang*

We appreciate the referee for her/his helpful and stimulating comments. We have prepared a revised manuscript in accord with the comments of the referee. The revised paragraphs and supplementary material have been inserted into the text. Some typographical errors have been corrected. Please see the revised manuscript. As to the specific responses in the revised paper, we would like to note the following modifications.

Reviewer comments:

(1) The previous studies on this topic, some cited in the manuscript, have already shown the beneficial effects of including ZnO nanowires on different fibers for filtration application. Thus, the authors should clarify the novelty of their work compared to the state-of-the-art literature.

  • The previous researches cited in the manuscript, the ZnO nanowire-based sensor comparing with the ZnO film sensor, shows 2 to 100 times enhanced photoconductivity on a limited area under 100 mm2. Here, ZnO NW based nanofiber membrane was performed 0.3 μm PM filtration at palm-size air filter scale, while the fabricated ZnO nanowire array was shown to retain photoelectric properties from the photoconductive results. Also, we added a detail explanation of the nanofiber and composite structures which were required to show importance using ZnO NW fabrication on the filter membrane and the novelty of this work.
  • The 3rd and 4th paragraphs of the "Introduction" section in the lines 45-49, and 53-55 were complemented as below, "Modulating of nanofiber surface (surface functionalization, and metal oxide combination), improves the content of active sites, which can be effective to absorb VOCs, sulfur dioxide and carbon monoxide. Souzandeh et al. presented a surface functionalization on the gelatin nano fabrics on a paper towel to capture PM and toxic gaseous molecules.", and "The nanofiber membrane filter, which is modulated nanofiber surface functionalization or the metal oxide combined structure, shows superior filtration efficiency and provides gas capture characteristics.".
  • And we added advantages to using ZnO nanowire structure on the PVDF nanofiber. "In addition, the wurtzite crystalline structure of the ZnO is beneficial to grow nanowire geometry at a low-temperature under 100℃. A low-temperature hydrothermal synthesis method is possible to fabricate hierarchical structure on a polymer membrane without thermal damages. [2]", in the lines 59-62.

(2) Authors should include the other studies, which have investigated similar materials and parameters.

  • As mentioned by referee, we added the related references with our topic as follows,
  • [20] Yang, K.; Yu, Z.; Yu, C.; Chen, H.; Pan, F.; An electrically renewable air filter with integrated 3D nanowire networks. Adv. Mater. Technol. 2019, 4, 1900101.
  • [21] Zhong, Z.; Xu, Z.; Sheng, T.; Yao, J.; Xing, W.; Wang, Y.; Unusual air filters with ultrahigh efficiency and antibacterial functionality enabled by ZnO nanorods. ACS Appl. Mater. Interfaces 2015, 7, 21538-21544.
  • [22] Pan, T.; Liu, J.; Deng, N.; Li, Z.; Wang, L.; Xia, Z.; Fan, J.; Liu, Y. ZnO Nanowires@ PVDF nanofiber membrane with superhydrophobicity for enhanced anti-wetting and anti-scaling properties in membrane distillation. J. Membr. Sci. 2021, 621, 118877

(3) The authors claimed that it is possible to improve the functionality of the filter through tuning different parameters. This is a too general claim, which should be supported by investigating some of the mentioned parameters.

  • We added further investigation analysis about pore size effect to pressure drop of a filter membrane. Pressure drop is mainly affected to the pore size of the nanofiber membrane, which can be controlled during the electrospinning process. We have included additional results in the manuscript and the supplementary material as below.
  • In the Manuscript, "Materials and Method" section in the lines 95-97, "The PVDF solution concentration is optimized to fabricate uniform nanofiber geometry for fabrication of the filter membrane with a low-pressure drop. (Figure S1 and S2 presented in the supplementary material)".
  • In the "Supplementary material", the paragraph was added as below, "The uniformity and diameter of the PVDF nanofiber is controlled by changing the concentration of the polymer solution. The viscosity of the 7 and 17 wt% PVDF solutions show the lower and the upper limit respectively, to fabricate nanofiber at the electrospinning. In the case of the concentration of 7 wt% PVDF solution condition, nanofiber geometry shows 120 nm average diameter, however, there are 67 beads over 2 μm size per 0.01 mm2 area. In contrast, concentration of 17 wt% PVDF solution condition, bead formation is not occurred due to stabilization of the polymer jet during the electrospinning process. High viscosity in the polymer jet increases the nanofiber diameter to 380 nm.".
  • And the paragraph, "In the electrospinning preparation step, PVDF solution concentration is controlled to 7, 9, 11, 13, 15, and 17 wt% in the acetone:DMF (7:3) mixture. The weight of nanofiber membrane is constantly controlled by changing the fabrication time based on the weight percent of the solute in the solution. The Figure S2 shows the pressure drop of the PVDF nanofiber membrane depending on the polymer concentrations.", is added in the "Supplementary material".

(4) It would be better to include the durability of the ZnO nanostructures on the PVDF surfaces (e.g., under vigorous washing conditions).

  • In the manuscript, a durability of the ZnO NWs-PVDF NF layer was described as shown in the lines 111-113, "After hydrothermal synthesis, the PVDF NF layer surface is washed with ethanol to clean out debris and drying at room temperature for 2 h.", which expression is enough to show the durability of the filter without its damage during the washing for dust elimination.

As mentioned by referee, the vigorous washing condition occurs surface damage to the ZnO NW layers.

(5) The authors should comment and better clarify the effects of the thickness/size of the applied ZnO layer/nanowires on the final applications.

• As mentioned by referee, the size effect of a filter should be considered to apply final application. In the "Supplementary material", we added figures and explanations to show the variation of the pressure drop depending on the nanofiber diameter of membrane.

Reviewer 2 Report

This paper reported a PVDF nanofiber membrane with ZnO nanowires. The ZnO nanowires-PVDF nanofiber layer filter shows high filtration efficiency for PM with 0.3 μm diameter. But the manuscript needs to be further polished, there are some parts on the discussion that need further clarification. My detail comments are listed as below.

  1. The third and fourth paragraph of the introduction need rewrite, the current logic is confused. The author should state the nanofiber, metallic oxide blenden NF and NF geometry clearly and tell the readers why you choose ZnO NWs.
  2. The title of this paper shows that the ZnO NWs are arrayed on PVDF nanofiber, so the author should characterize the photograph clearly.
  3. I suggest the part 3 (results) and part 4 (discussion) should be integrated to part 3 (results and discussion).

Author Response

Manuscript ID: applsci-1348520

"Hybrid Structure of ZnO Nanowire Array on PVDF Nanofiber Membrane/Nylon Mesh for Smart Filter: Photoconductive PM Filter" by Dong Hee Kang, Na Kyong Kim, Hyun Wook Kang*

We appreciate the referee for her/his helpful and stimulating comments. We have prepared a revised manuscript in accord with the comments of the referee. The revised paragraphs and supplementary material have been inserted into the text. Some typographical errors have been corrected. Please see the revised manuscript. As to the specific responses in the revised paper, we would like to note the following modifications.

Reviewer comments:

This paper reported a PVDF nanofiber membrane with ZnO nanowires. The ZnO nanowires-PVDF nanofiber layer filter shows high filtration efficiency for PM with 0.3 μm diameter. But the manuscript needs to be further polished, there are some parts on the discussion that need further clarification. My detail comments are listed as below.

(1) The third and fourth paragraph of the introduction need rewrite, the current logic is confused. The author should state the nanofiber, metallic oxide blended NF and NF geometry clearly and tell the readers why you choose ZnO NWs.

  • As referee mentioned, a detail explanation the nanofiber and composite structures was required to show novelty of this work and to reinforce the logic.
  • To reduce the confusion, the 3rd and 4th paragraphs of the “Introduction” section in the lines 45-49, and 53-55 were complemented as below, "Modulating of nanofiber surface (surface functionalization, and metal oxide combination), improves the content of active sites, which can be effective to absorb VOCs, sulfur dioxide and carbon monoxide. Souzandeh et al. presented a surface functionalization on the gelatin nano fabrics on a paper towel to capture PM and toxic gaseous molecules.", and "The nanofiber membrane filter, which is modulated nanofiber surface functionalization or the metal oxide combined structure, shows superior filtration efficiency and provides gas capture characteristics.".
  • And we added advantages to using ZnO nanowire structure on the PVDF nanofiber. "In addition, wurtzite crystalline structure of the ZnO is beneficial to grow nanowire geometry at a low-temperature under 100℃. A low-temperature hydrothermal synthesis method is possible to fabricate hierarchical structure on a polymer membrane without thermal damages. [2]", in the lines 59-62.

(2) The title of this paper shows that the ZnO NWs are arrayed on PVDF nanofiber, so the author should characterize the photograph clearly.

  • As mentioned by referee, we had been described the ZnO NW array on PVDF nanofiber including their scale and detail size information. "The ZnO NWs through the hydrothermal synthesis were grown vertical direction along the PVDF NF surface as in Figure 2d and e. The ZnO NWs on the PVDF NF surface show hierarchical structure with an average diameter of 1,158.5±482.2 nm." was described in the page 4, lines 134-137.

(3) I suggest the part 3 (results) and part 4 (discussion) should be integrated to part 3 (results and discussion).

  • As suggested by referee, the "Discussion" section was combined in the "Results and discussion" section in the lines 222-241.

Round 2

Reviewer 1 Report

The problems in the first draft have been addressed by the authors and the current version of the manuscript is acceptable.